# Comparison of Patterns of Skill Actions between Analog and Electronic Protectors in Taekwondo: A Log-Linear Analysis

**DOI:** 10.3390/ijerph17113927

**Published:** 2020-06-01

**Authors:** Eun-Hyung Cho, Han-Joo Eom, Se-Yong Jang

**Affiliations:** 1Korea Institute of Sport Science, Seoul 01794, Korea; ehcho@kspo.or.kr; 2College of Sport Science, Sungkyunkwan University, Suwon 16419, Korea; hjeom@skku.edu; 3Department of Physical Education, Gachon University, Seongnam 13120, Korea

**Keywords:** Electronic protector, general protector, log-linear analysis, martial arts, Taekwondo

## Abstract

The purpose of this study was to compare the patterns of skill actions executed during Taekwondo competitions when wearing and not wearing an electronic protector. To achieve this purpose, 110 matches from two university-level Taekwondo championships were taped and analyzed. The performance skills were composed of 18 detailed skills and grouped into five categories by considering kicks to the target area (chest or head/face). The data were organized in the form of a contingency table that demonstrated the relationship between grouping factors (skills, protectors, win–lose, and weight division). A log-linear analysis was carried out to investigate the effect of the grouping factors (IVs) on the skills (DV) using SPSS Statistics. The results obtained in the present study can be summarized as follows. First, the overall proportion of “points” called by the judge for the general protector (32.3%) was approximately 3.4 times that for the electronic protector (9.5%). Second, for the electronic protector, the proportions of kicks to the chest area were in the following order: Roundhouse kick (R-Kick) (44.7%), Pushing kick (P-kick) (19.3%), Turn kick (T-kick) (8.7%), and Double roundhouse kick (DR-kick) (7.6%). For the general protector, the order differed slightly, with T-kick and P-kick switched around with different proportions. Third, the proportion of kicks to the head/face was higher for the electronic protector (19.8%) than for the general protector (10.4%), and this difference was even more distinct when the light (−68kg) (33.5% (electronic) vs. 6.5% (general)) and heavy (+85kg) (1.4% (electronic) vs. 13.3% (general)) weight divisions were compared. Finally, the match status (win/lose) had no significant effect on the pattern of playing actions for both the protectors. The result from this study suggests that skill frequency of linear simple movement for activating electrical protector’s sensor is increased, while the one of rotational complex movement is decreased gradually. Additionally, headgear without sensors, such as for a hit movement to the face/head part, represent characteristics of increased attack skills to the facial area; these scores are provided through subjective judgement, and consequently changes in performance skills can occur.

## 1. Introduction

At the end of an official sports event, a result is produced. The outcomes of competitions are represented by certain numbers, which may be relative rankings, points, or scores produced by a certain scoring system. For example, in athletics and swimming, the relative ranking is determined by recording the distance or the time taken to complete a given task. In shooting and archery, the ranking is determined by the accumulated points assigned to the target. The scoring systems for team and net sports such as baseball, soccer, football, tennis and volleyball are all inherently designed to be highly objective. The decision-making process involved in the performance outcomes varies according to the unique characteristics of the sports competitions. The process of ranking an athlete’s performance is relatively simple, and the subjective judgement of game officials (i.2., movement) is not extensively involved in time- or distance-based events. The accuracy of a measured record is achieved in an objective manner using electronic devices. Currently, techniques based on high-speed cameras and videos are used to decide cases where doubt exists; such techniques make record measurement even more precise in terms of distinguishing the outcomes of actions in sports such as athletics, shooting, and archery. Additionally, in many team and net sports, an attempt has been made to increase the accuracy of a referee’s call by employing an IT system such as the Hawk-Eye officiating replay system. Such systems make it possible to provide a definitive answer in cases where determining what happened using the referee’s naked eye is difficult [1,2].

However, in other sports events such as gymnastics, figure skating, and combat sports (e.g., Taekwondo, Judo, and Wrestling), the performance itself is characterized by a series of continuous playing actions with interwoven complexities. The final outcome in these sports events in the results of cumulated points because point(s) are given by game referees during the middle of continuous playing actions. Therefore, the scoring process relies heavily on the subjective judgement of game officials and their naked eye [3,4].

Taekwondo is a globally known martial arts sport with practitioners in more than 206 countries. A Taekwondo player is required to have highly demanding physical and technical abilities to effectively perform a series of fast attack and counterattack actions against an opponent within a few seconds [5,6,7]. The subjective nature of the scoring system in Taekwondo competitions has been criticized and is often the subject of disputes and protests among coaches and players inside and/or outside the competition venue. These disputes have often attracted attention from the news and public media [8]. The World Taekwondo Federation (WTF) made an effort to resolve problems with the scoring system by introducing electronic scoring equipment with built-in wireless sensors in the uniform gear. This attempt was directed at increasing the accuracy and fairness of the scoring process by reducing possible intentional and/or unintentional controversial decisions by referees during competitions. The WTF organized the first testing event using an electronic protector at the International Taekwondo Championships held in Chuncheon, Korea, in March 2007. The electronic protector was used for the first time in an official competition at the 19^th^ Taekwondo World Championship in Copenhagen, Denmark, in October 2009 after a few negative outcomes at the 2008 Beijing Olympics because of controversial decisions resulting from the use of the general protector which is traditional analog uniform [9,10]. In an attempt to achieve better objectivity and transparency in scoring processes, for the 2012 London Olympics, the WTF introduced an upgraded electronic vest uniform with impact sensors in the scoring area of the chest and footpads lined with magnets. An additional electronic device—headgear—was featured for the first time in the 2016 Rio Olympics. The electronic protector was designed in such a way that a kick with a certain amount of force on the scoring area would activate the sensors and award point(s), which would be automatically and wirelessly transmitted to the judges [11]. Along with the electronic devices, the WTF made some rule changes, which first appeared at the 2012 London Olympics, such as an octagonal playing mat instead of a square one and a revamped point system. According to this system, a kick to the head/face earned three points instead of one and highly technical skills such as spinning kicks earned an extra point. The challenge card was also used for incidents when coaches disagreed with the machine’s scoring outcome. These modifications and rule changes appeared to be designed to encourage athletes to perform highly technical kicks and engage in faster and more aggressive fluid fights, which seem to have brought more excitement to competitions and greater transparency to the scoring process [12,13,14,15].

The electronic protector awards a point when a kick lands on a certain scoring area with a force that is just sufficient to activate the sensors. In contrast, when using the general analog protector, a kick should hit the scoring area, make a contact sound, and have a clear visual impact for the judges to recognize it as an effective attack. Since its appearance, the electronic protector has been the focus of research aimed at testing its reliability and accuracy [1,16] and assessing the sensitivity and consistency of the sensors for a variety of impact forces [17]. It was pointed out that certain areas on the chest guard seemed to provide easier scoring by default. It was also found that the sensors were activated even with a low impact force; thus, kicks with a high impact force were not needed unless athletes sought to disturb their opponents’ balance. Moon and Jung (2014) analyzed the skills exhibited during the women’s Taekwondo competition at the 2012 London Olympics and reported that the use of kicks to the head/face increased; such kicks, when successful, often determined match winners. Moreover, “front roundhouse kicks” to the chest area were the most frequently executed skill during competitions. Moon and Jung [18] also pointed out that the electronic scoring system was undesirably activated by a minor hit such as a “touch” impact and by a wired unique foot touch through unorthodox leg movements, suggesting the need for future improvements.

Regardless of how sophisticated electronic scoring becomes, it seems that it will never be perfect. Thus, a new updated version with new rules is expected to be introduced in the next editions of international competitions. In fact, the WTF announced a new scoring system in November 2016. The new technologies and rule changes have surely influenced the nature of competitions, and coaches and players should embrace new match tactics and strategies in response to the new point system. The purpose of this study was to analyze and compare the patterns of performance skills executed during international Taekwondo competitions with or without the electronic protector.

## 2. Materials and Methods

### 2.1. Data Collection

Table 1 two official competitions from the same category (university) with similar skill levels, which were held in Korea, were videotaped onsite by researchers, with the approval of the host organizations. The first event was the 2011 University Taekwondo Fall-Season Championship, where a general analog protector (hereafter called the general protector) was used. The second event was the 2012 World University Taekwondo Championship, where an electronic protector was used. These two events were authorized by the Korea Taekwondo Association and the World Taekwondo Federation. Each competition had 55 matches with four weight divisions.

### 2.2. Classification of Taekwondo Skills

Table 2 the sample matches were observed through video replay by all the authors (one of the authors is a former first-place winner of the World Championship), and skill executions were identified by mutual agreement among the observers. As listed in Table 2, 18 distinctive leg skills executed by the players were observed and classified into five main skill groups. These main skill groups were obtained from a literature review of related materials, and their use has been substantiated in Taekwondo research papers, books, and manuals published by Kukkiwon, World Taekwondo Federation, and Taekwondo Headquarters. In the process of itemizing the performance skills, the “Punching” skill was excluded because of a lack of frequency. Therefore, all the five skill groups analyzed in this study were “kick”-related Taekwondo skills. They were R-kick, DR-kick, T-kick, P-kick, and H-kick; the first four groups comprised of kicks to the chest area, and the last group comprised kicks to the head/face area. The full terms are listed in Table 2.

### 2.3. Data Input Program

A data input program was designed and developed using Excel macros (Figure 1). The program starts with inputting match information (use of an electronic protector (with or without) and a match number) and player information (name, uniform (blue or red), and weight division). Once the information is typed in, the input program is activated and four types of sequential “click” boxes are shown on the screen, namely, player identification, an attack target for the skill execution (body vs. head), the name of the executed skill, and the outcome of the skill execution (point vs. attempt).

The data input process was carried out when viewing a replayed match tape. A main person with former Taekwondo playing experience watched a replay and made a call regarding the attack target, name of the skill, and outcome. As the main person made a call, a trained assistant simultaneously clicked on the screen accordingly. A third observer (one of the authors) who sat in front of the video replay checked and confirmed the accuracy of both the judgment call and the click action on the screen. This data input process was repeated for all 110 matches, and data were accumulated and stored on the computer for further statistical analyses.

The sample matches were observed through video replay by all the authors (one of the authors is a former first-place winner of the World Championship), and skill executions were identified by mutual agreement among the observers. As listed in Table 2, 18 distinctive leg skills executed by the players were observed and classified into five main skill groups. These main skill groups were obtained from a literature review, and their use has been substantiated in Taekwondo research papers, books, and manuals published by Kukkiwon, World Taekwondo Federation, and Taekwondo Headquarters. In the process of itemizing the performance skills, the “Punching” skill was excluded because of a lack of frequency. Therefore, all the five skill groups analyzed in this study were “kick”-related Taekwondo skills. They were Roundhouse kick (R-kick), Double roundhouse kick (DR-kick), Turn kick (T-kick), Pushing kick (P-kick), and Head/face kick (H-kick); the first four groups comprised kicks to the chest area, and the last group comprised kicks to the head/face area. The full terms are listed in Table 2.

### 2.4. Statistical Analysis

Frequency data stored through the input data program were retrieved and transformed into SPSS data files for statistical analyses. Frequency data were grouped according to four factors: the five skill groups listed in Table 1, the types of protectors (general vs. electronic), the weight divisions (light (−68 kg.) vs. heavy (+80 kg.)), and the outcomes of a match (win vs. lose). Thus, a four-way multilevel (2 × 2 × 2 × 5) contingency table was formed, and a log-linear statistical method using the IBM SPSS Statistics 21.0 Program (SPSS Inc., USA) was applied to compare the proportions of the skill groups and investigate the effect of the types of protectors on the pattern of skill executions. Furthermore, an attempt was made to examine whether any differences existed in the pattern of the “protector by skill groups” interactions between the weight divisions and between the win and lose outcomes. A hierarchical log-linear analysis was selected in the model option using maximum likelihood (ML) chi-square statistics (G^2^). The G^2^ statistics in the program had two parts: one for partial association (i.e., a conditional test), which was calculated by controlling for the potential influence of the remaining variables, and the other for marginal association, which was obtained by collapsing or summing over the remaining variables without any adjustment [20,21]. Because our main interest is directed toward the effect of the type of protector on the cell distribution of skill categories, the G^2^ statistics for partial association are presented in the next section. As a post-hoc analysis, the odds ratios for each comparable cell for different grouping factors were calculated and presented to compare the differences in the proportions of the cell distribution between the general and electronic protectors.

## 3. Results

### 3.1. Results of Log-Linear Model on Four-Way Contigency Data

Table 3 presents the results of the log-linear analysis on the four-way full-model contingency table with ML chi-square statistics only for the partial association test. The magnitude of the G^2^ value for each term reflects the relative importance of a specific term in accounting for the distribution of the observed cell frequency. As Table 3 indicates, the main effect of each of the three factors was not statistically significant, indicating that the marginal frequencies in the outcome levels (win/lose), weight (light/heavy), and protector (electronic/general) were similar. However, the main effect of the skill factor, as expected, was significant (G^2^ = 2353.37, p < 0.001).

Table 4 as a result, an analysis of variance (ANOVA)-like table was formed that displays only those terms associated with testing for the effect of grouping variables on the distribution of skill execution and on the “protector by skill” association. Table 4 presents the results in the form of a multi-way analysis of variance. By treating Skill (S) as a dependent variable, which was accomplished by considering the given cell distributions, the effect of Protector (P) was found to be highly significant (G^2^ (4) + 139.44 for P × S term), indicating a meaningful difference in the cell distributions of skill execution between the general and electronic protectors. Furthermore, the effect of the weights (light vs. heavy) on the “P × S” association appeared to be considerably larger than that of the win/lose outcome, as shown in the magnitudes of G^2^ (414.48 vs. 58.49).

Table 3 presents a three-way (skill × protector × win/lose) contingency table. In the analysis of this table, a main aspect was to test whether the pattern of the “skill by protector” relationship had a distinctive difference depending on the matches won or lost. As the odds ratios show, the effect of the win/lose factor on the two-way (skill × protector) relationship was relatively small when compared to that of the weight factor. Moreover, this result could also be verified from the magnitude of the G^2^ values (58.9 vs. 414.8, respectively).

### 3.2. Comparisons of Cell Proportions for Two-Way (Skill by Protector) Contingency Table

The odds ratio, among various other statistics, is a statistic that can be easily used for research involving a post-hoc analysis that compares the relative cell proportions for a contingency table [22,23,24]. Table 5 presents the cell frequency (proportion) of a two-way (skill by protector) table. The total frequency in each protector was the sum of “attempt” and “point”. The frequency of “point” represents the number of executed skill actions that were called a point by the judge during the competitions, and the frequency of “attempt” represents the ones that did not receive a score. Furthermore, the odds ratio for each skill group presented on the right side of the table was calculated on the basis of the frequencies in each protector and the column total (i.e., [(niE/n.E)/(niG/n.G)] using the cross-product ratio formula [25](pp. 63). A graph of the cell proportions of the two protectors was presented as a visual reference.

As Table 5 indicates, the total frequencies of the electronic (2483) and general (2433) protectors were very similar, indicating that the players with an electronic protector and those without one executed a similar number of different skills within the limited time, which is also indicated by a non-significant result [G^2^(1) = 2.36, p = 0.2 = 125] in Table 3. Among the kick skills to the chest area summed for both the protectors, the R-kick was the most-frequently executed (50.5%), followed by P-kick (15.2%), T-kick (11%), and DR-kick (8%), which was the least-executed skill category. The proportion of kicks to the Face (H-kick) was 15.1%. However, the distributions of cell proportions were noticeably different between the two protectors. For example, as seen in the cell proportions, the values (reciprocal odds ratios) of R-kick, DR-kick, and T-kick for the general protector were 1.26, 1.12, and 1.56 times greater than those for the electronic protector was in the following order: R-kick, H-kick, P-kick, T-kick, and DR-kick. This ranking order was not the same for the general protector (T-kick was second and H-kick was fourth). Another noteworthy finding is that the overall proportion of “points” called by the judge for the general protector was 32.2% (783/2433) or 3.4 times greater than that of the electronic protector.

Table 3 as previously mentioned, the differences in the cell proportions of the skill factor between the general and electronic protectors were greater than usual when a third factor (weight division) was added to form a three-way contingency. The value of the G^2^ statistics increased from 139.44 to 414.48, indicating that the weight factor had a significant effect on the distributional pattern of the cell proportions. The cell proportions for the two protectors in the heavy weight division were very similar for R-kick, DR-kick, and T-kick, as indicated by the odds ratios being close to 1.0. However, the other two skills showed very different outcomes. For example, for P-kick, the cell proportion for the electronic protector was approximately twice that for the general protector, but for H-kick, the cell proportion for the general protector was considerably higher (9.1 times) than that for the electronic protector in the heavy weight divisions. In contrast, the distribution of cell proportions for the two protectors in the light weight divisions showed considerably different patterns from that in the heavy weight divisions. For example, the cell proportions for the general protector in the light weight division were considerably larger (in terms of the reciprocal odds ratios) than those for the electronic protector, for R-kick, DR-kick, and T-kick, except that the odds ratio for P-kick in the light weight division was similar to that in the heavy weight division. Furthermore, noteworthy differences existed between the light and heavy weights for H-kick when comparing the size of the cell proportions for the two protectors. 

### 3.3. Effect of Weight and Outcome Factors on Two-Way (Skill by Protector) Relationship

Table 6 H-kick was the second most frequently executed skill (33.5%) for the electronic protector. H-kick was also the last one (6.5%) for the general protector in the lightweight division but the order was reverse in the heavy weight division (1.4% and 13.3%, respectively). These differences between the light and heavy weight are well depicted in the figures presented on right side of the Table 6. 

Table 7 the cell proportion for R-kick, DR-kick, and T-kick was larger for the general protector than for the electronic protector and the cell proportion for the remaining two skills showed a smaller proportion for both the win and loss categories. Although some noticeable differences existed in the distributions of the cell proportion for T-kick and H-kick in the lose category, the overall cell proportion patterns for both the win and lose categories were very similar and their profiles resembled that of the two-way (skill × protector) relationship.

## 4. Discussion

Among the kicks to the chest area, R-kick was the most frequently executed scoring skill for both the protectors, and its frequency was considerably greater than that of any other skill (50.6% for the electronic protector and 69.2% for the general protector). The comparison of the ratio of attempts and the success rate of kicks to the head area between the two protectors also gave interesting results. For example, the total kick execution to the head area for the electronic protector was 19.8%, which was considerably higher than that for the general protector (10.4%). However, the success rate for the general protector (26.2%) was more than twice that for the electronic protector (12.6%). In general, these results indicated that, for scoring purposes, the players seemed to prefer performing simple and safe kick skills that required fast leg extension, regardless of the type of protector. However, with the electronic protector, athletes seemed to have made some changes in their scoring strategies by performing more kicks to the head area for extra points and more pushing-related kicks with a force that was just sufficient to activate the target sensors, instead of performing technical kicks that may require a “hit” impact on the target.

When a log-linear analysis is carried out on a multilevel contingency table with a number of categories in single factor, the main effect of each factor is generally not the point of interest. The main effect of a particular variable (or factor) being significant does not hold much importance other than the fact that the frequency in each category of the variable is not the same. Thus, the main focus of interest is on the interaction between two or more factors, or whether the cell frequency distribution (or proportion) in one variable is different or is influenced by the levels of another variable, which can be determined by a type of conditional probability test [26,27]. Therefore, in this study, our focus was directed at comparing the patterns of cell proportions in other grouping factors such as the weight division or the win–lose outcome, given the cell distributions of the skill factor.

The likelihood ratio of partial relevance can be calculated by controlling the potential influence of the remaining variables. Otherwise, the likelihood ratio can be obtained by sum or collapse of the states of the remaining variables without any adjustment. Therefore, while the results may not be directly associated, the partial relevance of a specific term in a multidimensional table may have different results from the periphery related to the state referred to in Simpson’s paradox [28,29,30]. The purpose of this study was to examine the related documents between technologies and protectors according to weight and protectors, so we did not pay attention to interpreting the meaning by examining all variables. According to weight and family type, so we did not pay attention to interpreting the meaning by examining all variables.

According to the results, there is a very close relationship between the type of protector and the performance skill used, and the highest likelihood ratio was found in P x S according to weight, with G^2^ = 414.48. In the end, there were many differences in P x S according to the light and heavy weight. However, the low likelihood ratio with P × S G^2^ = 58.49 according to the win or loss can be regarded as having little significance with the results. These results suggested that the performance skills executed by players during a competition were substantially influenced by the type of protector, and this difference was even greater when the heavy and light weight division players were compared.

Although the nature of whether these variables relationships are dependent on each other has not yet been accurately investigated, the results of this study can determine that they may be affected by the quality of the types of pitches used and the types of skills used in the study of the performance results of the athletes.

The analysis based on chi-square statistics is a test of independency between two variables. Although an obtained statistic indicates the statistical significance of the association between two variables based on a criterion level, it does not clearly show where the dependency occurred or the strength of the relationship between two variables.

For further research, we would like to analyze the electronic and general protector data used in this study by applying the latest updated competition rules, and to generalize the results by conducting research that applies to national-level athletes.

## 5. Conclusions

This study compared the patterns of skill actions in Taekwondo competitions with and without an electronic protector. Two competitions from the same category (university) with similar levels of playing characteristics were selected, and all matches were taped for video-replay analyses. A log-linear statistical method was applied to analyze the generated sample data. The results of the analyses can be summarized as follows: First, the overall proportion of “points” called by the judge for the general protector (32.2%) was approximately 3.4 times larger than that for the electronic protector (9.5%). Second, the proportions of kicks to the chest area for the electronic protector were in the following order: 44.7% (R-kick), 19.3% (P-kick), 8.7% (T-kick), and 7.6% (DR-kick). For the general protector, the order was slightly different, with T-kick and P-kick being switched around with different proportions: R-kick (56.5%), T-kick (13.5%), P-kick (11.1%), and DR-kick (8.6%). Second, the proportion of kicks to the head/face was higher for the electronic protector (19.8%) than for the general protector (10.4%). This difference was even more distinct when the light weight division (33.5% (electronic) vs. 6.5% (general)) and the heavy weight division (1.4% (electronic) vs. 13.3% (general)) were compared. Fourth, the match status (win/lose) had no significant effect on the pattern of playing actions for both the protectors.

## Figures and Tables

**Figure 1 ijerph-17-03927-f001:**
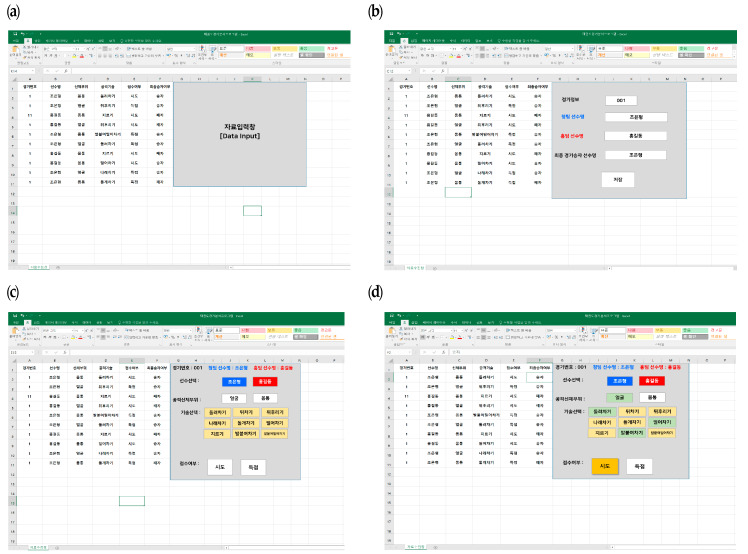
Data input program for Taekwondo. (**a**) Match Information; (**b**) Player Identification; (**c**) Attack target for the skill execution; (**d**) Outcome of the skill execution

**Table 1 ijerph-17-03927-t001:** Data source.

Protector	Competition	Weight Division	Number of Matches	Total
General Protector	2011 Fall Korean University Taekwondo Championship	58 kg 68 kg 80 kg +80 kg	13 14 14 14	55
Electronic protector	2012 World University Taekwondo Championship	58 kg 68 kg 80 kg +80 kg	13 14 14 14	55
Total				110

**Table 2 ijerph-17-03927-t002:** Classification of Taekwondo Skills.

Target	Skill Group	Skills (18)
Chest	1. Roundhouse kick (1)	Roundhouse kick (Dollyeochagi)
	2. Double roundhouse kick (2)	Double roundhouse kick (Naraechagi) + Running kick (Balbuchyeochagi)
	3. Turn kick (2)	Turn kick (Dolgaechagi) + Back kick (Dwichagi)
	4. Pushing kick (3)	Pushing kick (Mireochagi) + Running pushing kick (Balbutyeo-mireochagi)
Head	5. Head/Face kick (10)	Roundhouse kick (face), Back kick (face), Double roundhouse kick (face), Tornado kick (face), Pushing kick (face), Downward kick, Balbuchyeo downward kick, Balbuchyeo kick (face), Balbuchyeo pushing kick (face), Back spin kick

Note: The classification of Taekwondo skills was following by Taekwondo textbook [19].

**Table 3 ijerph-17-03927-t003:** Decomposition of G^2^ partial associations for entire frequency tables.

Effect	*df*	G^2^ Chi Square	Significance (p)
Win/lose (V)	1	0.980	0.322
Weight (W)	1	0.129	0.720
Protector (P)	1	2.36	0.125
Skill (S)	4	2353.37	<0.001
V × W	1	392.82	<0.001
V × P	1	20.79	<0.001
W × P	1	96.49	<0.001
V × S	4	32.74	<0.001
W × S	4	201.13	<0.001
P × S	4	139.44	<0.001
V × W × P	1	11.99	<0.001
V × W × S	4	58.09	<0.001
V × P × S	4	58.49	<0.001
W × P × S	4	414.48	<0.001
V × W × P × S	4	9.43	0.051
Total	39		

**Table 4 ijerph-17-03927-t004:** ANOVA-like table for partial association G^2^, with (P × S) being treated as dependent measures.

Variable		Partial Test		
Independent	Dependent	G^2^	*df*	*p*
	S	2353.37	4	<0.001
	P × S	139.44	4	<0.001
Weight (W)	P × S	414.48	4	<0.001
Win/lose (V)	P × S	58.49	4	<0.001
W × V	P × S	9.43	4	0.051

Note: *df* = degree of freedom. *p* = probability.

**Table 5 ijerph-17-03927-t005:** Observed frequencies for attempt and point (%) and odd ratios for protector by skill contingency table.

Skills	Protector									
	Electronic					General				
	Attempt	Point	P%	Sub Total	Odds Ratio	Attempt	Point	P%	Sub Total	E/G Odds Ratio
R-kick	990	119	10.7	1109	0.869	833	542	39.4	1375	0.79
	44.0	50.6		44.7		50.5	69.2		56.5	
DR-kick	174	15	7.9	189	1.22	147	62	30.0	209	0.89
	7.7	6.4		7.6		8.9	7.9		8.6	
T-kick	200	15	7.0	215	1.39	257	71	21.6	328	0.64
	8.9	6.4		8.7		15.6	9.1		13.5	
P-kick	455	24	5.0	479	1.98	227	42	15.6	269	1.74
	20.2	10.2		19.3		13.8	5.4		11.1	
H-kick	429	62	12.6	491	0.72	186	66	26.2	252	1.91
	19.1	26.4		19.8		11.3	8.4		10.4	
Total	2248	235	9.5	2483		1650	783	32.2	2433	

**Table 6 ijerph-17-03927-t006:** Observed frequencies (proportions) and odd ratios for skill by protector by weight contingency table.

Frequency	Light			Heavy		
	E-p	G-p	Odds Ratio	E-p	G-p	Odds Ratio
R-kick	520 36.6	633 60.0	0.61	589 55.4	742 53.8	1.03
DR-kick	81 5.7	77 7.3	0.77	108 10.2	132 9.6	1.06
T-kick	95 6.7	158 15.0	0.47	120 11.3	170 12.3	0.91
P-kick	247 17.4	118 11.2	1.56	232 21.8	151 11.0	1.99
H-kick	476 33.5	69 6.5	5.16	15 1.4	183 13.3	0.11
Total	1419 100.0	1055 100.0		1064 100.0	1378 100.0	

**Table 7 ijerph-17-03927-t007:** Observed frequencies (proportions) and odd ratios for skill by protector by win/lose contingency table.

Frequency	Win	Lose
	E-p	G-p	Odds Ratio	E-p	G-p	Odds Ratio
R-kick	54543.1	68757.8	0.745	56446.3	68855.3	0.837
DR-kick	1068.4	1048.8	0.957	836.8	1058.4	0.806
T-kick	1149.0	12810.8	0.836	1018.3	20016.1	0.515
P-kick	25219.9	13010.9	1.82	22718.6	13911.2	1.67
H-kick	24719.5	13911.7	1.67	24420.0	1139.1	2.21
Total	1264100.0	1188100.0		1219100.0	1245100.0

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
