# Peer review of "Comparison of Patterns of Skill Actions between Analog and Electronic Protectors in Taekwondo: A Log-Linear Analysis"

_ijerph, 2020, doi:10.3390/ijerph17113927_

Round 1
Reviewer 1 Report
The authors examine the skill patterns in university taekwondo tournaments in relation to the type of protector, and link them to the dependent variables weight class, technique success and win/ loss.
The methodology contains some weaknesses. There is a terminological ambiguity, which sometimes speaks of analog, traditional and general protectors as opposed to the electronic protector. This inaccuracy begins in abstract line 2. Since the data were collected in 2011/ 2012, it is not said which iteration of the electronic protector is used, as on the occasion of the London Olympics 2012 changes in rules and protector equipment were made. Furthermore, the authors write that electronic headgear was not used until 2016. This means that the fifth skill category (kicks to the head) can in principle not be dependent between electronic and analog protectors. No explanation is given as to which of the four weight categories examined are categorized light and heavy and to what extent this grouping is statistically permissible at all. It would also have been methodologically wise to have the tournament with electronic protectors judged by the researchers themselves applying the usual analog protector referee guidelines. In this case, there would be indications why the scoring points of analog protectors are more than three times higher than that of electronic protectors.
The actual research question is also imprecise: Why do the authors even assume that the usage of fight techniques changes depending on the respective protectors and that there is a correlation? Do the fighters adapt their techniques to the way the protectors work? Maybe several interviews with participants would shed some light onto this question. At the very least, assumptions concering these correlations should be listed, outlined or discussed within the framework of the study.
The interpretation of the cross combination of different characteristics (weight division, loss/ win, attempt/ point) does not seem to be sustainable enough, regarding the low density of the available data.
Formally, there are also some errors to be found:
- The figure 1 is rather uninteresting for the reader, because it only visualizes the input of the data.
- On page 4 and 5 there are completely identical paragraphs.
- On page 9 there is talk of an F-Kick, which otherwise does not appear in the plain text.
- Page 11 says "our focus was our focus was”.
Author Response
I would like to thank you for reviewing this research.
There are some changes based on your comments, Thanks.
- There is a terminological ambiguity, which sometimes speaks of analog, traditional and general protectors as opposed to the electronic protector.
Response: All changes are highlighted in red
- This inaccuracy begins in abstract line 2. Since the data were collected in 2011/ 2012, it is not said which iteration of the electronic protector is used, as on the occasion of the London Olympics 2012 changes in rules and protector equipment were made. Furthermore, the authors write that electronic headgear was not used until 2016. This means that the fifth skill category (kicks to the head) can in principle not be dependent between electronic and analog protectors.
Response: The data
- No explanation is given as to which of the four weight categories examined are categorized light and heavy and to what extent this grouping is statistically permissible at all.
Response: There was in text.
- The figure 1 is rather uninteresting for the reader, because it only visualizes the input of the data.
Response: Although it did not looks interesting, it is necessary to show the process of using data.
- On page 4 and 5 there are completely identical paragraphs.
Response: All changes are highlighted in red
- On page 9 there is talk of an F-Kick, which otherwise does not appear in the plain text.
Response: All changes are highlighted in red
- Page 11 says "our focus was our focus was”.
Response :All changes are highlighted in red
Thanks,

Reviewer 2 Report
Comments and suggestions for Authors are in the attachment.

Author Response
I would like to thank you for reviewing this research with interesting.
There are some changes based on your comments, Thanks.
- participant: age, gender, body height, body mass, sports career, ethical approval
Response: The data based on video record, so there was no information about participants.
- kind of games (elimination, final) recorded during the championships
The game was tournament from the round of 16 to final game
Tables general
all tables should be formatted in the same way (mainly horizontal lines must be corrected),
description of the table must be before not after them,
Table 1. Data source
information about the total number of games is not necessary, the number of games per
competition should be reported.
Table 2. Taekwondo skill classification
insert skills shortcuts in the bracket e.g. Roundhouse kik (R-kick), Double roundhouse kick
(DR-kik), etc.
Tables 3-4.
delete the first row,
use the same statistics order in columns and use the symbols of statistics e.g. G2, df, and p.
Tables 5-7.
the first column must be named skills not frequency,
use the same frequency and proportion notation e.g. 990 (44.0),
column 6 should be deleted in table 5,
insert the same shortcuts in all tables e.g. EP and GP.
Figures
Figure 1. Data input program for Taekwondo.
the figure is illegible. It should be prepared in the English language or delete.
Figure 2-4.
use the same shortcuts as in the tables.
Writing style must be improved:
Incorrect citation style:
.
Incomplete writing style:
R-kick (44.7%), P-kick (19.3%), T-kick (8.7), and DR- ,
A log-linear analysis was carried out to investigate the effect of the grouping factors (IVs)
.
Wrong writing style:
a similar number of different skills within the limited time, which is also indicated by a
non-significant result [G2(1) = 2.36, p = .2=125] in table 3,
2(4) + 139.44 for P x S
term.
Response: all changes are highlighted in red

Reviewer 3 Report
General Comments
Current research intituled “Comparison of Patterns of Skill Actions between Analog and Electronic Protectors in Taekwondo: A Log-Linear Analysis” has good quality and potential to be well accepted by technical and research community.
Major specific comments
Abstract
. Withdraw using IBM...21.0
. What is “R-kick, P-kick, T-kick, and DR-kick”? Specify.
. Rephrase: “...the lightweight divisions were compared.”
Introduction
. Review paragraph lengths...they are unbalanced.
. Page1 and 2: There is too much information about other sports disciplines. The authors should review these first paragraphs and be objective.
. Is there any hypothesis?
. The authors should clearly mention the main contributions of the present study in the last Introduction paragraph.
Materials and Methods
. “..., including the first author, with approval...” Withdraw including the first author.
. “ ... and the World Taekwondo Federation.” Withdraw respectively.
. Pg 5. “...five main n skill groups, which were ...Headquarters.” Withdraw “of related materials”
. Pg 5: “They were R-kick ... area” If possible, authors should add a Figure with panels that illustrate these skills.
. Pg 5: Figure 1 is not well visible.
. Pg 5. Figure 1 title should clarify what is 1 (a panel), 2 (b panel) in detail.
Results
If authors have considered the Results and Discussion section, this separation should be clearly visible in the manuscript, which is not the present case.
. Pg 6: “...were very similar” Withdraw very.
. Pg 6: “..highly significant (G...).” Withdraw highly.
. Pg 6,7 and 8: The tables titles should be written in detail and explaining all abbreviations.
. Pg 7: “...appeared to be considerably larger than...” Appeared? The authors should use statistics to describe the results. It does not seem well-understood use the term appeared with such a detailed statistics applied.
. Pg 6: “The odds ratio,...”. This sentence is not needed. It should be included in the Materials and Methods section.
. Pg 8: Figure 2 title should be described in detail Furthermore, this Figure should be presented in the manuscript text before its appearance.
. Pg 9 and 10: Tables 6 and 7, Figures 3 and 4 should be presented in the manuscript text before its appearance and both titles should be described in detail.
. Pg 9: “...contingency table 3.”
. Pg 9 and 10: “ ... the figures presented on the right side...” Withdraw this information.
Discussion
. Pg 11: The first paragraph is confusing. It should be presented just for the main study findings. Limitations, procedures, suggestions should be placed in the proper manuscript section.
. Pg 11: “...our focus was our focus...”
. Pg 11: “.Therefore, the purpose of this study was to ...” it should be a reminder in the first paragraph, but not indispensable.
. Pg 11: “ ... so we did not pay attention...variables.” It should be discussed as a limitation or future suggestion in the last paragraph.
. Pg 11: “ ... Although nature ... athletes.” Rewrite clearly and consider the paragraph length balance along with the manuscript.
. Pg 11: “The analysis based ...variables. This paragraph is too short and is out of the manuscript reading context flow.
. pg 11: “Among the kicks performed to the chest area...” this paragraph should be the first in the Discussion section
Conclusions
. Pg 12: This section is not the Abstract. Consider rewrite pointing out what are the main conclusions. .Pg 12: “Why is this important...judges? I really don’t understand this sentence. Consider reviewing.
Author Response
I would like to thank you for reviewing this research with interesting.
There are some changes based on your comments, Thanks.
- Abstract
. Withdraw using IBM...21.0
. What is “R-kick, P-kick, T-kick, and DR-kick”? Specify.
. Rephrase: “...the lightweight divisions were compared.”
Response: all changes are highlighted in red
Introduction
- . Review paragraph lengths...they are unbalanced.
. Page1 and 2: There is too much information about other sports disciplines. The authors should review these first paragraphs and be objective.
Response: we would like to talk about taekwondo skills, so it was described widely.
. Is there any hypothesis?
Response: No, we did not set up the hypothesis.
- Materials and Methods
. “..., including the first author, with approval...” Withdraw including the first author.
. “ ... and the World Taekwondo Federation.” Withdraw respectively.
. Pg 5. “...five main n skill groups, which were ...Headquarters.” Withdraw “of related materials”
. Pg 5: “They were R-kick ... area” If possible, authors should add a Figure with panels that illustrate these skills.
. Pg 5: Figure 1 is not well visible.
. Pg 5. Figure 1 title should clarify what is 1 (a panel), 2 (b panel) in detail.
Response: all changes are highlighted in red
- Results
. Pg 6: “...were very similar” Withdraw very.
. Pg 6: “..highly significant (G...).” Withdraw highly.
. Pg 6,7 and 8: The tables titles should be written in detail and explaining all abbreviations.
. Pg 7: “...appeared to be considerably larger than...” Appeared? The authors should use statistics to describe the results. It does not seem well-understood use the term appeared with such a detailed statistics applied.
. Pg 6: “The odds ratio,...”. This sentence is not needed. It should be included in the Materials and Methods section.
. Pg 8: Figure 2 title should be described in detail Furthermore, this Figure should be presented in the manuscript text before its appearance.
. Pg 9 and 10: Tables 6 and 7, Figures 3 and 4 should be presented in the manuscript text before its appearance and both titles should be described in detail.
. Pg 9: “...contingency table 3.”
. Pg 9 and 10: “ ... the figures presented on the right side...” Withdraw this information.
Response: all changes are highlighted in red
- Discussion
. Pg 11: The first paragraph is confusing. It should be presented just for the main study findings. Limitations, procedures, suggestions should be placed in the proper manuscript section.
. Pg 11: “...our focus was our focus...”
. Pg 11: “.Therefore, the purpose of this study was to ...” it should be a reminder in the first paragraph, but not indispensable.
. Pg 11: “ ... so we did not pay attention...variables.” It should be discussed as a limitation or future suggestion in the last paragraph.
. Pg 11: “ ... Although nature ... athletes.” Rewrite clearly and consider the paragraph length balance along with the manuscript.
. Pg 11: “The analysis based ...variables. This paragraph is too short and is out of the manuscript reading context flow.
. pg 11: “Among the kicks performed to the chest area...” this paragraph should be the first in the Discussion section
Response: all changes are highlighted in red
- Conclusions
. Pg 12: This section is not the Abstract. Consider rewrite pointing out what are the main conclusions. - .Pg 12: “Why is this important...judges? I really don’t understand this sentence. Consider reviewing.
Response: it was withdrawn
Thanks,

Reviewer 4 Report
Thank you for this paper. I believe that this is a very relevant topic for Taekwondo.
However, I am not prepared to completely review this paper. I was reading page 4 and page 5 is essentially a repeat of page 4. This is an error that MUST be corrected before even submitting a paper. Once corrections such as this have been made, I will be willing to read the whole paper and review. If the authors do not have time to correct major errors like this before submission, I do not have time to read the whole paper.
Additional comments. Overall, the English is very good. The first paragraph does not have to be as long as it is. Give brief examples of objective sports, and then the problems with Taekwondo. This can be shorter.
Please give a reference for your categories of kicks. There is probably a book or other reference that has all of these skills listed. Please give a cited reference here so people not familiar with Taekowndo will be able to research these if needed.
Sometimes you write games and sometimes matches. Please be consistent and only use one term. What is most common? I suspect matches is more appropriate, but I am not an expert.
Table 1 needs to be clearer and corrected.
You wrote that a "person" watched the video and gave points for hits. Please change this to judge or another appropriate term.
How did you control reliability of judgement of the videos? Can you comment on this and add it to the methods? How many "judges" were used, is this enough to ensure scientific objectivity and reliability?
I will read the entire paper once the major error I cited is corrected.
Author Response
I would like to thank you for reviewing this research with interesting.
There are some changes based on your comments, Thanks.
The purpose of this study is to analyze statistically the differences in the frequency of taekwondo techniques used in general and electronic protector.
The electronic protector is recognized by the strength of the sensor. Therefore, the frequency of use of Taekwondo's unique kicking techniques has decreased, and it is to show that the frequency of use of simple techniques which can represent scores well is increasing.
- Sorry there was a repeated sentences in page 4 and 5. Changes in red.
Please give a reference for your categories of kicks. There is probably a book or other reference that has all of these skills listed. Please give a cited reference here so people not familiar with Taekowndo will be able to research these if needed.
- The classification of Taekwondo skill was following by Kukkiwon Teakwondo text book.
Sometimes you write games and sometimes matches. Please be consistent and only use one term. What is most common? I suspect matches is more appropriate, but I am not an expert.
- the Term changed game to match
How did you control reliability of judgement of the videos? Can you comment on this and add it to the methods? How many "judges" were used, is this enough to ensure scientific objectivity and reliability?
-The results obtained through the agreement of three experts, including the main research team, were evaluated to increase reliability.

Round 2
Reviewer 3 Report
General Comments
Current research intituled “Comparison of patterns of skill actions between analog and electronic protectors in Taekwondo: a log-linear analysis”(IJERPH-770628-peer-review-v2) has originality, rationality, and completeness of research problem. Overall, the manuscript presents good tables and figures. The authors have done a good job answering previous questions and improving the manuscript considering previous suggestions. However, an additional effort is requested for manuscript quality improvement and publication following the minor points presented below.
Major specific comments
Abstract
Page 1: ... R-kick..., P-kick..., DR-kick...” The authors should clarify what means R, P, T, DR before the kick. Page 1: Consider revising space between numbers and units, where it is appropriated.
Introduction
Page 1: (i.2, judges) is it correct?
Materials and Methods
Page 4: Table 2 title is incomplete.
Page 4: “The full terms are listed in Table 2.” The Table 2 presentation in the manuscript must precede Table 2.
Page 5: “The full terms ... repeated.
Page 5: Figure 1 has poor quality. Explain in the title what means 1, 2, 3, and 4 and I suggest you use a, b, c, and d panels.
Results
Page 7: The full Table 4 presentation in the manuscript must precede Table 4.
Page 8: There is no reason to include a note. Review in the Table title.
Page 8: Where Figure 2, Table 6, and Figure 3 presentation is?
Page 9: Table 6 title is incomplete.
Page 9: Figure 3, The title is incomplete.
Page 9: “...contingency (Table 3).”
Page 10: Where Table 7and Figure 4 is?
Page 10: “...in the figures presented on the right side of table 6.” I can ́t see this layout in the manuscript. It should be reviewed.
Page 10: Table 7 title is incomplete.
Page 10: “ Table 3 presents a three-way (...) contingency table. This sentence is too far from Table 3. Is it correctly depicted on this page?
Page 11: “(58.9 vs. 414.48, respectively).
Page 11: “...on the right side of Table 7”? Consider reviewing.
Discussion
Page 11: “Therefore, in this study, our focus...” Next paragraph... “Therefore, the purpose of this study. Consider rewrite.
Page 12: The last paragraph from the Discussion section seems a limitation paragraph. The authors should add strong points controlled along the study to minimize the impact of methodological limitations. The authors should also add a future study suggestion sentence.
Conclusions
Page 12: Consider rewrite concisely this section highlighting just the main finding, without repeating text.
Author Response
Thank you for the detail, I revised it as your comment.
Comments and responses in attached file.
Thanks,

Reviewer 4 Report
Thank you for your corrections. I still have some issues with this paper, I will attach my comments. I have highlighted my areas of concern in your PDF, and have included another PDF with my specific comments to the areas highlighted. I believe that I know what you are trying to say, but I do not feel that your purpose is clear and I am not sure if you really show why this is important to Taekwondo. I believe that it is, but you have not clearly demonstrated this for me. I want to see the practical relevance of this paper.

Author Response
Thank you for details,
comments and responses in attached file.
Thanks,
